# Material Design and Optimisation of Electrochemical Li-Ion Storage Properties of Ternary Silicon Oxycarbide/Graphite/Tin Nanocomposites

**DOI:** 10.3390/nano12030410

**Published:** 2022-01-26

**Authors:** Dominik Knozowski, Pradeep Vallachira Warriam Sasikumar, Piotr Madajski, Gurdial Blugan, Maria Gazda, Natalia Kovalska, Monika Wilamowska-Zawłocka

**Affiliations:** 1Department of Energy Conversion and Storage, Faculty of Chemistry, Gdańsk University of Technology, Narutowicza 11/12, 80-233 Gdańsk, Poland; dominik.knozowski@pg.edu.pl; 2Laboratory for High Performance Ceramics, Empa, Swiss Federal Laboratories for Materials Science & Technology, CH-8600 Dübendorf, Switzerland; pradeep.variyar@gmail.com (P.V.W.S.); Gurdial.Blugan@empa.ch (G.B.); natalia.kovalska@empa.ch (N.K.); 3Faculty of Chemistry, Nicolaus Copernicus University in Torun, 87-100 Toruń, Poland; piotr.madajski@doktorant.umk.pl; 4Department of Solid State Physics, Faculty of Applied Physics and Mathematics, Gdańsk University of Technology, Narutowicza 11/12, 80-233 Gdańsk, Poland; margazda@pg.edu.pl

**Keywords:** silicon oxycarbide, tin nanoparticles, Li-ion battery, ternary composites, graphite

## Abstract

In this work, we present the characterization and electrochemical performance of various ternary silicon oxycarbide/graphite/tin (SiOC/C/Sn) nanocomposites as anodes for lithium-ion batteries. In binary SiOC/Sn composites, tin nanoparticles may be produced in situ via carbothermal reduction of SnO_2_ to metallic Sn, which consumes free carbon from the SiOC ceramic phase, thereby limiting the carbon content in the final ceramic nanocomposite. Therefore, to avoid drawbacks with carbon depletion, we used graphite as a substitute during the synthesis of precursors. The ternary composites were synthesized from liquid precursors and flake graphite using the ultrasound-assisted hydrosilylation method and pyrolysis at 1000 °C in an Ar atmosphere. The role of the graphitic component is to ensure good electric conductivity and the softness of the material, which are crucial for long term stability during alloying–dealloying processes. The presented approach allows us to increase the content of the tin precursor from 40 wt.% to 60 wt.% without losing the electrochemical stability of the final material. The charge/discharge capacity (at 372 mA g^−1^ current rate) of the tailored SiOC/C/Sn composite is about 100 mAh g^−1^ higher compared with that of the binary SiOC/Sn composite. The ternary composites, however, are more sensitive to high current rates (above 372 mA g^−1^) compared to the binary one because of the presence of graphitic carbon.

## 1. Introduction

The exceptional storage capabilities of tin, such as the high gravimetric capacity of 994 mAh g^−1^ and the relatively low lithiation potential of 0.6 V [1,2,3], make it an appealing anodic material for lithium-ion batteries. Indeed, much effort has been devoted to developing stable tin-based anodes. The number one obstacle to overcome is their high-volume expansion of around 260% during the lithium insertion process. This leads to pulverization of electrodes, i.e., material rupture and disintegration upon cyclic volume changes, resulting in poor electrochemical stability over continuous charge/discharge cycles [4]. The most common way to prevent pulverization is the reduction of Sn particle size to the nanometre scale [5,6] and their homogenous distribution in a matrix to prevent their agglomeration. Moreover, the matrix has to be able to withstand the strain created upon volume expansion during cycling. Multiple materials have been tested as a potential matrix for Sn nanoparticles (NPs), including metal-based [7,8], carbon-based [9,10], and ceramic-based materials [11,12,13,14]. From these three types of matrices, polymer-derived ceramics (PDCs) deserve special attention because of their useful and unique features. From the PDC family, silicon oxycarbides (SiOCs) have been widely studied as anodes for Li-ion batteries, as they exhibit a high capacity of ~600 mAh g^−1^, good stability, and high rate capability [15,16]. SiOCs are produced by pyrolysis of preceramic precursors, such as polysiloxanes, polysilsesquioxanes, and polycarbosiloxanes [17]. They are amorphous structures consisting of short-range ordered silicon–oxygen–carbon tetrahedra (SiO_x_C_4-x_, x = 0–4) surrounded by a free carbon phase. Their composition and structure can be easily tailored through multiple synthesis routes [18,19,20,21,22,23,24,25]. Moreover, SiOCs exhibit good mechanical properties and low volume expansion upon lithiation, which makes them great candidates for electrochemically active matrices for Li-alloying nanoparticles. It has been shown that a large amount of free carbon phase dispersed in the ceramic matrix stabilizes nanoparticles over extended charge–discharge cycles [11].

One of the ways to incorporate Sn NPs into the ceramic matrix is by mixing a preceramic polymer with a liquid tin precursor in the pre-pyrolysis stage. To obtain a homogenous mixture, assuring the uniform dispersion of Sn particles in the matrix as well as their nanometre size, the polarity of a tin precursor has to match the polarity of the preceramic polymer [13]. The formation of tin nanoparticles takes place upon pyrolysis in a two-step process. Firstly, tin oxide is formed at low temperatures (>200 degrees), followed by their carbothermal reduction into metallic tin (>700 degrees), consuming the carbon present in the ceramic matrix. This process leads to uniformly distributed Sn NPs, but at the same time, it also reduces the carbon content in the final material. The carbon phase in SiOC-based composites is essential for ensuring the appropriate softness of the material and electrical conductivity [12,15,26]. Insufficient carbon content within the material causes quick performance fading, and for the best performing material, the tin content within the ceramic matrix is limited to a maximum of ~30 wt.% [13].

The addition of an extra source of carbon during preceramic synthesis is expected to be beneficial for increasing the tin content in the final ceramic nanocomposites. The carbon content in SiOCs may be increased by admixing heavy aromatic hydrocarbons, such as pitch [27,28], divinylbenzene (DVB) [29,30], and polystyrene [31], or by adding carbonaceous materials, such as graphene [32,33], carbon nanotubes [34], and graphite [35,36,37,38]. The incorporation of graphite is an interesting option for a couple of reasons. Graphite is a very cheap, abundant, and non-toxic material. Moreover, it exhibits a stable electrochemical response and low-volume expansion upon lithium insertion, and it has one of the lowest lithiation potentials of 0.05 V [3]. Graphite was already successfully applied in combination with PDCs [35,37,38,39]. It is worth noting that the addition of graphite to the ceramic material may induce different electrochemical properties, depending on the composition of the preceramic polymer and the polymer-to-graphite ratio [35]. A capacity increase, a lower lithiation potential, a higher first-cycle Coulombic efficiency, and a better rate capability were observed for various SiOC/graphite composites, which have been mostly attributed to the improved conductivity, the creation of new lithium storage sites [37,38], and the appropriate ratio between ordered (graphitic) and disordered carbons [35]. Graphite was also reported to be a good component for Sn-based nanocomposites due to its low volume expansion and good electrical conductivity [5]. Graphite reduces charge transfer resistance and lowers overall polarization [40]. In addition, the graphite/tin composite exhibits a remarkably improved capacity at low temperatures in comparison with pure graphitic anodes [41].

In this work, we combined graphite and SiOC to achieve an exceptional matrix for tin nanoparticles and increase carbon content, which is a crucial component for ensuring good conductivity. The idea behind the addition of graphite was also to increase the tin content in the final material compared with that of the pure carbon-rich ceramic matrix. The ternary silicon oxycarbide/graphite/tin composites were synthesised using a poly(methylhydrosiloxane)–divinylbenzene mixture as a preceramic precursor, tin (II) octoate as a tin source, and graphite flakes. The choice of the preceramic polymer and the tin source was based on the good miscibility of these components as reported by Dubey et al. [13]. The main focus of this work was on the influence of graphite with regard to the progress of carbothermal reduction processes as well as the composition, microstructure, and electrochemical behaviour of the final composites.

## 2. Materials and Methods

### Synthesis of SiOC/C/Sn Composites

Starting materials: Polymethylhydrosiloxane (PMHS, MW ≈ 1900), divinylbenzene (DVB, technical grade, 80%), graphite (flakes, 20 μm), tin octoate (Sn(Oct)_2_, 92.5–100.0%, Sigma-Aldrich), Karstedt’s catalyst (2% of Pt element in xylene), and acetone (≥99.5%) were purchased from Sigma Aldrich, Buchs, Switzerland. All materials were used as received.

Synthesis of the active materials: The composites were synthesised using the ultrasound-enhanced hydrosilylation method. First, 5 μL of Karstedt’s catalyst was added to 2 g of DVB. Then, after 2 min of intense stirring, 2 g of PMHS, 2.5 g of acetone, and different amounts of graphite and tin octoate were added to the mixture. After a couple of minutes of stirring, the suspension was treated with an ultrasonic probe sonicator for 3–5 min to obtain a solid gel. The gels of various compositions were dried for two days at 80 °C to complete crosslinking. Finally, the green bodies were pyrolyzed using the following programme: heating to 250 °C with a rate of 100 °C h^−1^, dwelling at 250 °C for 2 h, heating to 1000 °C with a rate of 150 °C h^−1^, dwelling at 1000 °C for 1 h, cooling to room temperature with a rate of 60 °C h^−1^. The compositions of the preceramic blends are presented in Table 1. The matrix was composed either of the pure ceramic (SiOC) or the ceramic mixed with graphite (SiOC:C_x_, where the “x” value corresponds to the mass fraction of graphite replacing the preceramic polymer). The tin precursor content is denoted as Sn-y%, where “y” corresponds to the weight percentage of the tin precursor with regard to the mass of the whole preceramic polymeric blend.

Characterization techniques: Raman analysis was conducted using a micro-Raman spectrometer (InVia, Renishaw, Wotton-under-Edge, UK) equipped with an Ar ion laser (514 nm) within 100–3200 cm^−1^. The analysis of obtained spectra was performed using OriginPro2016 software after background subtraction with custom settings. The modes were fitted according to [42], using Lorentzian fitting for D4, D1, G, and D2 peaks, and Gaussian fitting for the D3 peak. XRD reflexes were collected using the EDAX, RTEM model SN9577. The morphology of obtained materials was characterised using a scanning electron microscope (Tescan VEGA, Tescan, Brno, Czech Republic) and a transmission electron microscope (FEI, G2 F20X-Twin 200 kV, FEG), while the surface elemental distribution was assessed using a Phenom XL Scanning Electron Microscope (ThermoFisher Scientific, Warsaw, Poland) equipped with an energy dispersive X-ray (EDX) microanalyser. Brunauer–Emmett–Teller (BET) surface area analysis was performed on a BET analyser (SA 3100, Beckman Coulter, Brea, California, USA). Magic angle spinning nuclear magnetic resonance (MAS NMR) measurements for silicon ^29^Si NMR were conducted on the Bruker Avance Ultrashield 500 MHz spectrometer (Billerica, MA, U.S) and the Avance Ultrashield 500 MHz spectrometer (Waltham, MA, USA). The parameters were as follows: single pulse sequence, ^29^Si frequency: 139.11 MHz, π/8 pulse length: 2.5 ms, recycle delay: 100 s, 1 k scans, external secondary reference: DSS. Under air flow, 3.2 mm zirconia rotors filled with samples were spun at 8 kHz. The spectra were fitted using OriginPro2016 software, assuming they consisted of Lorenz type peak components.

Electrochemical measurements: Electrodes for electrochemical testing were prepared using the doctor blade technique. First, the finely grounded active material (85 wt.%) was ball milled with carbon black (7.5 wt.%, Super P, TIMCAL, Bodio, Switzerland) at 350 rpm. Then, carboxymethyl cellulose (7.5 wt.%, CMC, SUNROSE MAC 500LC, Nippon Paper Group) solution in water was added to the active material/carbon black mixture, and then all components were further ball milled for 1.5 h. The obtained slurries were coated on Cu foil, and the layers were then dried overnight in a vacuum oven at 120 °C. Swagelok^®^ type cells were prepared using 1 M LiPF_6_ in EC:DMC (1:1) (ethylene carbonate: dimethyl carbonate, BASF, battery grade, >98%, TCI Chemicals (Portland, OR, USA), a soaked glass microfiber separator (MN GF-2, Macherey-Nagel GmbH & Co. KG, Düren, Germany, thickness 45 µm), and lithium foil (Sigma Aldrich, Schnelldorf, Germany) as the counter electrode. Electrochemical measurements were conducted on a multichannel battery interface (Atlas 0961, Atlas-Sollich, Rębiechowo, Poland) and/or on the Biologic Potentiostat SP200 (BioLogic Science Instruments, Seyssinet-Pariset, France). The samples were measured after a waiting time of ~20 h. The galvanostatic charge–discharge cycles were performed using the constant current–constant voltage protocol (CCCV) in the potential range of 0.005 V–1.5 V with a 15 min potential hold after the lithiation step. Cyclic voltammetry (CV) measurements were conducted with a scan rate of 0.1 mV s^−1^ within the same potential range.

## 3. Results and Discussion

The carbon phase in SiOC significantly influences their properties as matrices for alloying nanoparticles [11] as well as their electrochemical performance. As we described in our recent work [35], not only is the amount of carbon important but also its structure. We showed that the incorporation of the ordered carbon phase in the form of small graphitic flakes in the SiOC structure is beneficial for electrochemical activity towards lithium ions and cycling stability.

A thorough analysis of SiOC/Sn and SiOC/C/Sn composites by means of Raman spectroscopy helps to correlate the carbon structure with the composites’ properties.

The Raman spectra of all investigated materials (Figure 1a) exhibited D and G modes at ~1340 cm^−1^ and ~1580 cm^−1^, respectively, characteristic of carbonaceous materials. For further analysis, these modes were deconvoluted into five peaks [42], as presented in Figure 1b–f, Appendix A, and Appendix A and Appendix A (in the Appendix A). Pure SiOC exhibits a typical response with a strong D1 peak (*I*_D1_ = 0.950), related to the disordered forms of carbon, and about half-size G and D2 peak intensities (*I_G_* = 0.441 and *I_D_*_2_ = 0.379), corresponding to the ideal and damaged graphitic lattice, respectively [42,43]. The addition of the tin precursor to SiOC (SiOC/Sn-40% sample) led to minor changes in the carbon structure compared to pure SiOC—slightly less intense D1 and D2 bands. On the other hand, replacing part of the ceramic with graphite (in the case of ternary SiOC/C/Sn composites) caused a significant rise in the G band intensity, which is an expected effect for the introduction of ordered carbon networks in the material. An interesting effect is noted when one compares samples with the same amount of graphite but of different tin content. The SiOC:C_0.2_/Sn-60% sample exhibits a lower *I*_D1_/*I*_G_ ratio than the SiOC:C_0.2_/Sn-40% sample (0.481 vs. 0.551, respectively), suggesting a higher fraction of ordered carbon in sample SiOC:C_0.2_/Sn-40%. This indicates that tin oxide consumes mainly the disordered free carbon phase during the carbothermal reduction process.

The structure was further investigated by means of XRD analysis (Figure 2). Pure SiOC (Figure 2 black curve) displayed only a noisy signal with a halo at ~22⁰, which is typical for these types of amorphous materials [44]. All ternary composites exhibited a series of sharp peaks coming from graphite and tin. The peaks at 26.2° and 55° confirm the presence of a hexagonal structure of graphite within the composite materials. Other graphitic peaks are missing, probably because of hindrance by the ceramic phase. The remaining peaks of ternary composites correspond to *β*-tin (the PDF card Sn-ref_00-004-0673 is presented in Appendix A in Appendix A). No signals related to SnO_2_ were observed, suggesting complete carbothermal reduction of SnO_2_ during pyrolysis. The relative intensities between graphitic and *β*-Sn depend on the composition, with the graphitic peak more pronounced for the graphite-rich SiOC:C_0.2_/Sn-40% sample and the *β*-Sn peaks the most intense for the SiOC:C_0.1/_Sn-60% material.

The investigations of morphology, structure, and composition were conducted using SEM-EDX techniques. Figure 3a,b show the morphology of SiOC and SiOC:C_0.2_/Sn-40%, respectively. The pure SiOC powder consisted exclusively of slate rock-like particles of various sizes and shapes. By contrast, tin-containing composites exhibited a large fraction of the crumble-like particles of a coarse texture with plenty of white precipitants and numerous cavities and darker regions with sizes of several microns marked by red circles in Figure 3b (more SEM images for deeper insight are presented in Appendix A and Appendix A in Appendix A). The EDX elemental maps (Figure 3c–e) show two distinctive regions: the zones composed mainly of carbon, which can be assigned to the graphite flakes, and the remaining part containing silicon, oxygen, carbon, and tin, which corresponds to the ceramic part. One may notice that the tin nanoparticles are accumulated exclusively in the ceramic part, where a low signal coming from carbon is observed. This result suggests that the free carbon phase from the ceramic part is preferably consumed during the carbothermal reduction process, which is also reflected in the Raman spectra analysis. Such behaviour may be explained by a better mixing of the tin source with the preceramic precursor than with graphite (liquid–liquid in contrast to liquid–solid mixing) as well as a very high activation energy for the carbothermal reduction of tin with graphite [45,46]. This assumption is further supported by the unique morphology of the ceramic region with the cavities and pores coming from the release of CO_2_ during the carbothermal reduction. Pure SiOC ceramic is a dense material (specific surface area (SSA) of 3.18 m^2^ g^−1^), whereas the SiOC/Sn-40% composite exhibits SSA of 118.1 m^2^ g^−1^ (results of BET analysis are shown in Appendix A and Appendix A in Appendix A). In the case of the composite materials, the SSA of ternary composites is smaller than that of the binary composite because of the lower ceramic content, which here is partially replaced by graphite. Moreover, comparing the SSA of the composites with the same amount of ceramic and graphite but different tin contents, one may observe higher SSA for SiOC:C_0.2_/Sn-40% (106.9 m^2^ g-^1^) than for SiOC:C_0.2_/Sn-60% (55.1 m^2^ g^−1^). This may result from the more intensive carbothermal reduction process in the ceramic phase, in which the release of CO_2_ creates and enlarges porosity, and then the enlarged pores may collapse, leading to the decreased SSA. Furthermore, the SiOC:C_0.1_/Sn-60% composite exhibits a higher SSA (94.6 m^2^ g^−1^) than SiOC:C_0.2_/Sn-60% because of the higher ceramic content and thus a less intensive CO_2_ release during the carbothermal reduction. These results further support the hypothesis that graphite does not take part in the carbothermal reduction process.

A deeper insight into the microstructure of the ternary composites was provided by transmission electron microscopy. TEM images of the binary SiOC/Sn-40% and the ternary SiOC:C_0.2_/Sn-60% composite are shown in Figure 4a,b, respectively. The microstructure of SiOC/Sn-40% consists of tin nanoparticles, represented by black dots, immersed in a featureless amorphous structure of SiOC [47,48]. In comparison, the SiOC:C_0.2_/Sn-60% composite reveals additional parallel smudges typical for graphite [49], which interpenetrate the ceramic component. In both samples, tin nanoparticles are uniformly distributed within the ceramic part. The average size of tin nanoparticles seems to depend on the amount of the ceramic part in the composite, as presented in Appendix A in the Appendix A. Figure 5a presents the HAADF image of the ternary SiOC:C_0.2_/Sn-60% composite with clearly separated ceramic and graphitic phases. This image confirms that the tin nanoparticles accumulate mainly in the ceramic part (Figure 5b), while in the graphitic phase, there are only a few randomly distributed tin nanoparticles (Figure 5c). The EDX elemental compositions of the selected regions of Figure 5a) are presented in Appendix A in the Appendix A. The differences between the ternary composites of various tin content are presented in Appendix A in the Appendix A. It is shown that the samples with a lower content of the tin precursor show much smaller tin accumulation and greater distances between each tin nanoparticle than samples where 60 wt.% of tin precursor was added. These results confirm that carbothermal reduction occurs mainly in the ceramic phase.

The influence of the carbothermal reduction on the ceramic microstructure was tracked through the ^29^Si solid-state NMR measurements. The pure ceramic sample (Figure 6a) exhibited peaks corresponding to various SiO_x_C_y_ mixed bonds tetrahedra, with plenty of Si–C bonds present [50]. On the other hand, all the ternary SiOC/C/Sn composites (Figure 6b–d)) revealed only one peak at around −114 ppm, which corresponds to SiO_4_ units [51,52]. No peaks indicating Si–C bonds were detected. The presence of exclusively SiO_4_ units indicates the depletion of carbon atoms bound to Si, which was explained by oxidation of C and Si by SnO_2_ during its reduction to metallic Sn [14]. These results suggest that during carbothermal reduction, not only the free carbon phase but also carbon from the mixed bonds of silicon tetrahedra are preferably consumed compared with graphite.

The electrochemical activity of the composite materials was investigated by means of cycling voltammetry and galvanostatic charge–discharge techniques. The CV curves present a general electrochemical pattern similar for all the investigated ternary composites, as presented in Figure 7a,b and Appendix A in Appendix A. The first cathodic sweep differs from the following cycles. In the first lithiation process, a long shoulder between 1.4 V and 0.66 V emerges, disappearing in subsequent cycles, which can be attributed to the creation of a solid-electrolyte interface (SEI) on all of the three components of the composites [19,53,54,55]. Then, at lower potentials, a broad feature at about 0.46 V appears, which is related to the formation of different Li_x_Sn_y_ alloys, mainly LiSn. In the following cycles, this peak splits into two peaks with maxima at 0.32 V and 0.61 V, which correspond to Li_7_Sn_2_ and Li_2_Sn_5_ phases, respectively [55,56,57]. In the potential range between 0.36 V and 0.005 V, multiple reactions occur, which can be attributed to the creation of high-lithium tin alloys and Li^+^ intercalation into the graphite and ceramic phases [53,58,59,60]. During the anodic sweep at low potentials, three peaks at 0.1, 0.17, and 0.24 V appear, which come from a gradual deinsertion of lithium ions from graphite [53,59]. The following peaks in the range between 0.3 and 0.9 V correspond to a gradual dealloying of Li_x_Sn_y_ [57]. A small broad peak at approximately 1.1 V, fading with the following cycles, can be attributed to partial oxidation of ~1 nm size Sn nanoparticles [61,62]. The cyclic voltammetry curves confirm the electrochemical activity of all components of the composites and indicate the creation of both high intercalation graphite stages (LiC_6_) and high-lithium tin compounds. The electrochemical activity of the ternary composites of different compositions is compared in Figure 7b. The activity of graphite is the lowest for the SiOC:C_0.1_/Sn-60% sample, which is a direct consequence of having the lowest graphite content among all the samples. In addition, the tin alloying currents are the smallest for the sample with the lowest tin content, namely SiOC:C_0.2_/Sn-40%. These results suggest that the electrochemical response of each component is related to the amount of that component in the composite material.

Figure 8 presents voltage profiles for the first and second cycles for ternary composite materials in the 0.005–1.5 V potential range. In the first lithiation process, there are no distinct plateaux, which corresponds well to the first cathodic polarization on the CV curves (Figure 7a and Appendix A in Appendix A), where no distinct redox peaks are observed. In the second and the following cycles, multiple plateaux are observed, indicating the creation of Sn_x_Li_y_ alloys and stepwise Li^+^ intercalation into graphite. During delithiation steps the redox activities of graphite (Figure 8a inset) and tin (Figure 8b) towards lithium ions are clearly depicted. The sloping shape of charge–discharge curves is typical for the ceramic component. The results confirm the electrochemical activity of all the composite components. The initial lithiation capacity for obtained materials varies between 1000–1300 mAh g^−1^, and it drops by approximately half of the value over the first few cycles. The first delithiation capacity equals 439, 507, and 515 mAh g^−1^ for the SiOC:C_0.2_/Sn-40%, SiOC:C_0.2_/Sn-60%, and SiOC:C_0.1_/Sn-60% samples respectively, which gives first cycle efficiency of around 45% in the 0.005 V–1.5 V potential range (57–63% in the 0.005 V–3 V potential range), as presented in Table 2. Such high capacity loss during the first cycle is mostly related to the creation of complex SEI graphite, tin, and ceramic phases and to irreversible lithium bonding with the ceramic phase [16,63]. The SEI creation process is additionally enhanced by a high specific surface area, which increases the contact between the active material and electrolyte and thus promotes higher lithium-ion consumption [64].

Figure 9a presents cyclic stability measurements of ternary composites within the 0.005 V–3 V potential range. After the initial cycles, the capacity of tin-rich samples, i.e., SiOC:C_0.2_/Sn-60% and SiOC:C_0.1_/Sn-60%, is around 610 mAh g^−1^, while the capacity of SiOC:C_0.2_/Sn-40% stays around 480 mAh g^−1^. This indicates that the tin nanoparticles are the major lithium storage sites within composites, and increasing tin content indeed leads to a higher capacity. In subsequent cycles, the capacity of SiOC:C_0.1_/Sn-60% gradually fades, while for other materials, it remains rather stable. Fast capacity fading for SiOC/Sn composites was previously reported as the result of insufficient carbon content within the material due to the carbothermal reduction process [12]. This problem is more pronounced in the composites in which the tin precursor was used in quantities higher than 40 wt.% with regard to a mass reaction mixture [13]. In the case of the SiOC:C_0.1_/Sn-60% composite, the stability over 100 cycles is much worse than that for the SiOC:C_0.2_/Sn-60% (52.7 vs. 86.5%). The results show that graphite, although not protecting the free carbon phase from the consumption in the carbothermal reduction process, can in an appropriate amount provide the required softness and electrical conductivity for good stability over extended cycling. SEM images of the electrode layers before and after cycling are presented in Appendix A (in Appendix A). The morphology of the electrode layers of the composites with a lower number of tin nanoparticles (SiOC/Sn-40% and SiOC:C_0.2_/Sn-40%) does not change after extended cycling tests (Appendix A), whereas the SiOC:C_0.2_/Sn-60%) exhibit some cracks (Appendix A, which may be attributed to the higher Sn concentration and associated volume changes upon alloying as the tin nanoparticles are accumulated mainly on the ceramic part.

The XRD analysis of cycled samples are presented in Appendix A (in Appendix A). The diffractograms are quite complex and reveal the presence of graphite, *β*-Sn (components of the composites), Li_2_CO_3_, Sn_3_O_4_ (components typical for SEI on tin containing electrodes) [65,66,67], Li_2_Sn_5_ (one of the lowest lithiation states of Li-Sn alloys), and Cu (current collector). The XRD patterns of the cycled binary and ternary composites are similar. A quantitative analysis would be ambiguous because of the overlapping of the XRD reflexes and strong peaks coming from the copper current collector.

To give some overview, we compared our ternary composites to the state-of-the-art SiOC/Sn-40%, which was reported as the best performing SiOC/Sn composite [13]. The capacity of SiOC/Sn-40% is around 520 mAh g^−1^ with a stability of 90.1% over 100 cycles. It appears that the capacity of SiOC/Sn-40% is lower than the capacity of the SiOC:C_0.2_/Sn-60% sample but higher than that of SiOC:C_0.2_/Sn-40%, while the stability is similar to the stability of these samples. These results indicate that the capacity strongly depends on the composition in a way that the presence of tin increases the capacity of composites, while graphite decreases it, which can be expected taking into account their theoretical capacity values. However, graphite can be a good component improving the ability of the host matrix to host higher tin content, which in turn can help increase the overall capacity of a ternary composite compared to the binary one.

The rate capability measurements within the 5 mV–1.5 V potential range are presented in Figure 9b. At lower current rates (C/5 and C/2 in respect to graphite) the SiOC:C_0.2_/Sn-60% composite exhibits the highest capacity values. However, with increasing polarization currents (above 1C–372 mA g^−1^) capacity values of all the ternary composites decrease more rapidly than for the binary SiOC/Sn-40% composite. Moreover, after returning to the low polarization currents, the capacities of the ternary composite do not increase as in the case of the binary composite. These results show that the composites with graphite are sensitive to high currents. The pure graphitic electrode also exhibits capacity fading at currents higher than 1C due to uneven SEI formation and lithium plating [68,69]. On the other hand, SiOC ceramics are known as stable materials at high current rates [24,70,71].

## 4. Conclusions

In this work, we reported a novel silicon oxycarbide–graphite–tin nanoparticle composite for lithium-ion batteries. Samples were produced using flake graphite and liquid preceramic and tin precursors via an ultrasound–enhanced hydrosilylation reaction. Employing high power probe sonication ensures the homogeneous mixing of all components and accelerates the cross-linking process. Final ternary composites were obtained through pyrolysis, which transformed the preceramic polymer into carbon-rich silicon oxycarbide and tin precursor into the tin nanoparticles via carbothermal reduction. This resulted in homogeneous composites with tin nanoparticles uniformly distributed mainly in the ceramic phase.

We observed that during carbothermal reduction of the tin precursor, carbon coming from the free carbon phase and mixed bonds silicon tetrahedra is preferably consumed over graphite. This is related to better mixing of the tin precursor with the preceramic polymer, as well as the high activation energy for the SnO_2_–graphite reaction. In our system, graphite provided appropriate conductivity and softness for stable alloying–dealloying of tin nanoparticles upon multiple cycles and also actively intercalated/deintercalated lithium ions. Replacing part of the ceramic matrix with graphite enabled an increase in the tin content with regard to literature reports, without deteriorating electrochemical stability upon prolonged charge-discharge cycles. Our best performing sample, SiOC:C_0.2_/Sn-60%, delivers a capacity about 100 mAh g^−1^ higher than the state-of-the-art SiOC/Sn-40% measured at the same conditions, while the stability at 1C was roughly the same. However, ternary composites are sensitive to high polarization currents (above 1C) because of the graphitic component.

## Figures and Tables

**Figure 1 nanomaterials-12-00410-f001:**
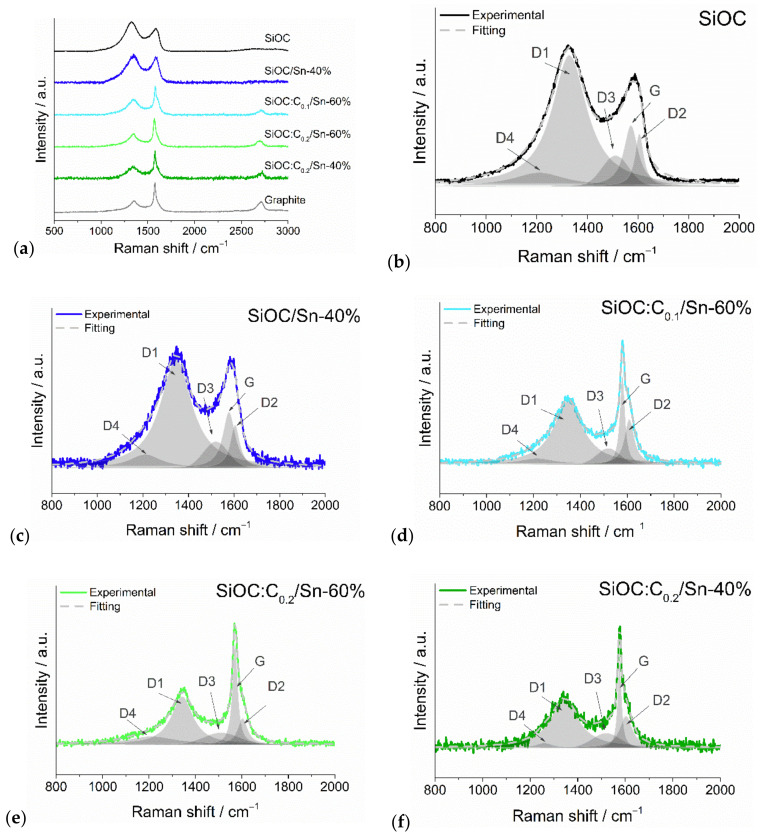
(**a**) Raman spectra of SiOC, graphite and composite materials after background subtraction with custom settings, fitting results for (**b**) pure SiOC, (**c**) SiOC/Sn-40%, (**d**) SiOC:C_0.1_/Sn-60%, (**e**) SiOC:C_0.2_/Sn-60%, and (**f**) SiOC:C_0.2_/Sn-40%.

**Figure 2 nanomaterials-12-00410-f002:**
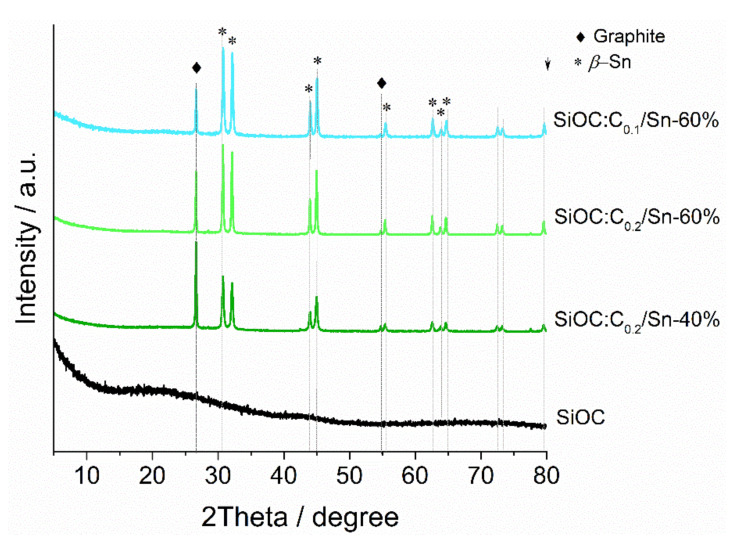
Diffractograms of SiOC and ternary composite materials.

**Figure 3 nanomaterials-12-00410-f003:**
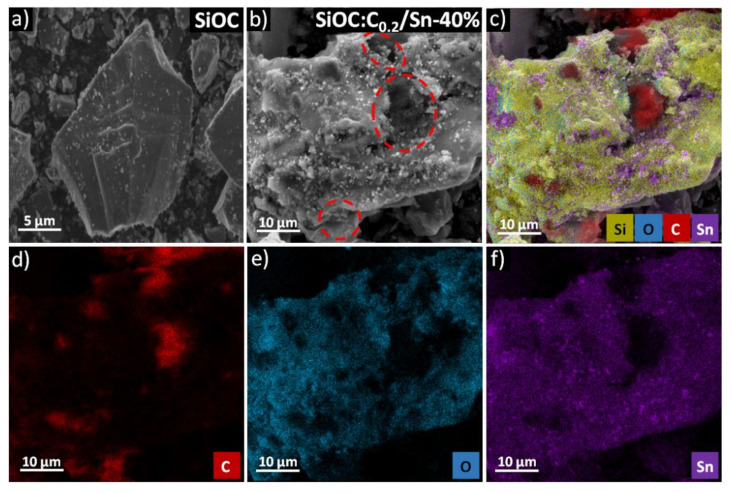
SEM image of (**a**) SiOC and (**b**) SiOC:C_0.2_/Sn-40% composite; (**c**–**f**) X-ray elemental maps of silicon, carbon, oxygen, and tin of the SiOC:C_0.2_/Sn-40% composite.

**Figure 4 nanomaterials-12-00410-f004:**
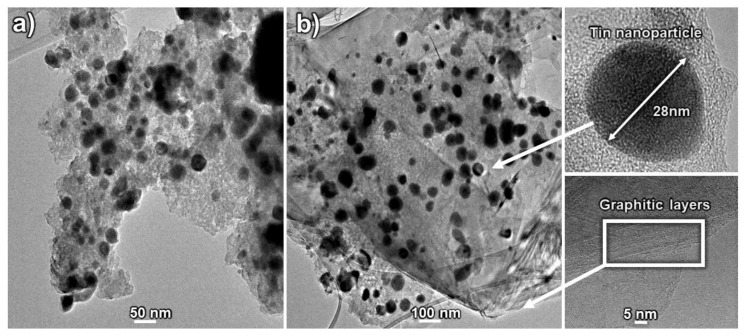
TEM pictures of (**a**) SiOC/Sn-40% and (**b**) SiOC:C_0.2_/Sn-60% composites.

**Figure 5 nanomaterials-12-00410-f005:**
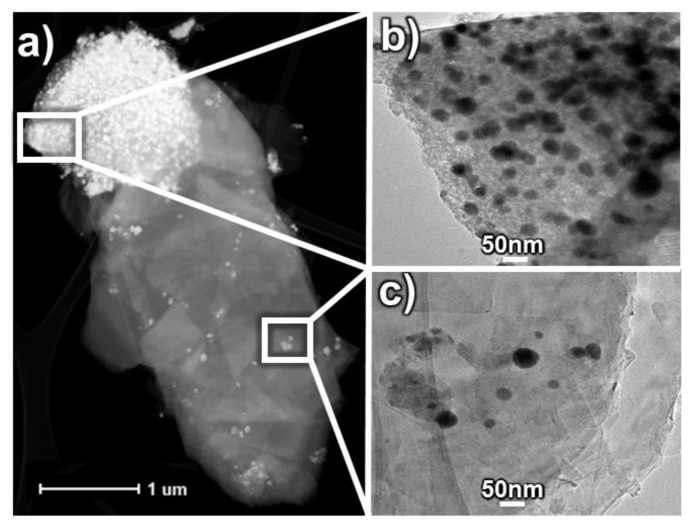
(**a**) HAADF image of a SiOC:C_0.2_/Sn-60% particle with separated ceramic and graphitic phases; TEM images of (**b**) ceramic phase and (**c**) graphitic phase.

**Figure 6 nanomaterials-12-00410-f006:**
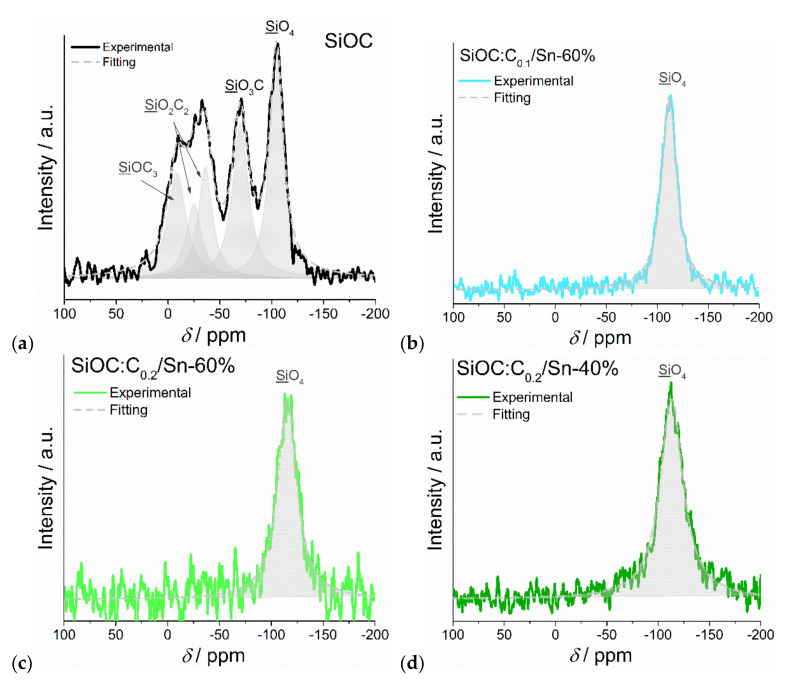
^29^Si NMR spectra for (**a**) SiOC (**b**) SiOC:C_0.1_/Sn-60%, (**c**) SiOC:C_0.2_/Sn-60% and (**d**) SiOC:C_0.2_/Sn-40%.

**Figure 7 nanomaterials-12-00410-f007:**
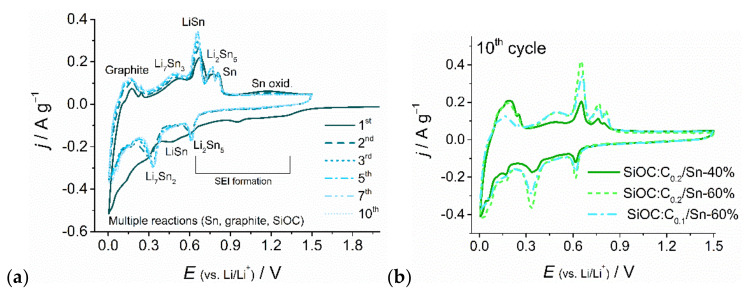
(**a**) Cyclic voltammetry measurements of selected cycles of SiOC:C_0.1_/Sn-60% composite, and (**b**) comparison of the 10^th^ cycle of the ternary composites.

**Figure 8 nanomaterials-12-00410-f008:**
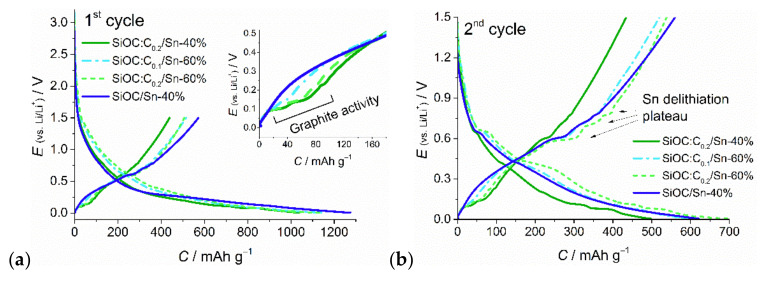
Voltage profiles of the ternary composites: (**a**) 1st cycle, (**b**) 2nd cycle.

**Figure 9 nanomaterials-12-00410-f009:**
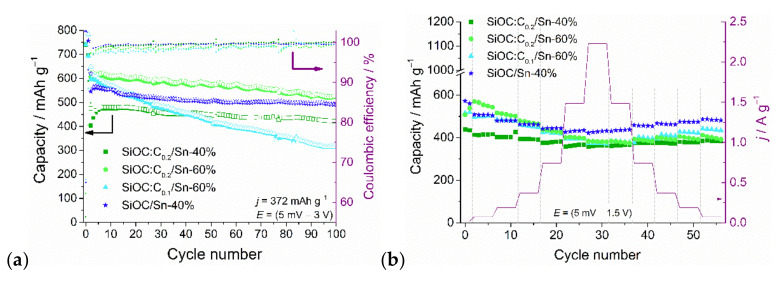
(**a**) Cycling stability at 372 mA g^−1^ and the corresponding Coulombic efficiencies; (**b**) rate capability of the ternary SiOC/C/Sn composites.

**Table 1 nanomaterials-12-00410-t001:** Compositions of the preceramic blends.

Composite	Tin Octoate	Matrix	
		Graphite	PMHS/DVB; 1:1 w/w Ratio
SiOC	-	-	5 g
SiOC/Sn-40%	3.33 g	-	5 g
SiOC:C_0.2_/Sn-40%	3.33 g	1 g	4 g
SiOC:C_0.2_/Sn-60%	7.5 g	1 g	4 g
SiOC:C_0.1_/Sn-60%	7.5 g	0.5 g	4.5 g

**Table 2 nanomaterials-12-00410-t002:** Selected electrochemical results. Coulombic efficiencies *η* within 0.005 V–1.5 V and 0.005 V–3 V potential regions were calculated by dividing the first lithiation capacity by the first delithiation capacity obtained in certain potential regions.

Material	1st Cycle C_irrev_/mAh g^−1^	1st Cycle C_rev_/mAh g^−1^	*η* (0.005 V–1.5 V)/%	*η* (0.005 V–3 V) /%
SiOC:C_0.2_/Sn-40%	600	439	42	56.5
SiOC:C_0.2_/Sn-60%	623	507	45	62
SiOC:C_0.1_/Sn-60%	625	515	45	63
SiOC/Sn-40%	699	572	45	65

## Data Availability

The data presented in this study are available on request from the corresponding author.

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
