# Peer review of "Material Design and Optimisation of Electrochemical Li-Ion Storage Properties of Ternary Silicon Oxycarbide/Graphite/Tin Nanocomposites"

_nanomaterials, 2022, doi:10.3390/nano12030410_

Round 1

Reviewer 1 Report

The article deals with the actual topic of creating new materials for lithium-ion batteries. Improving their performance and capacity characteristics is essential for modern materials science.

The article is undoubtedly suitable for publication in the journal Nanomaterials.

There are a few comments:

  1. When preparing the studied materials used platinum catalysts, neither in the text of the article nor in the supplementary it is not specified how to get rid of the catalyst. Please comment.
  2. For clarity, authors can add to the Supplementary microphotographs of a larger scale to evaluate the morphology of the particles (in addition to Fig. 3).
  3. Have diffraction and morphology analyses been performed on the material after electrochemical testing? This analysis could also profitably complement the above work.

Author Response

The article deals with the actual topic of creating new materials for lithium-ion batteries. Improving their performance and capacity characteristics is essential for modern materials science.

The article is undoubtedly suitable for publication in the journal Nanomaterials.

Thank you for your valuable comments concerning our manuscript. We corrected the manuscript according to your suggestions.

There are a few comments:

  1. When preparing the studied materials used platinum catalysts, neither in the text of the article nor in the supplementary it is not specified how to get rid of the catalyst. Please comment.

The catalyst was not removed but it was added only in negligible amount (only 5 µL of catalyst containing 2% of Pt was used per 8.3 – 12.5 g of substrate mixtures).

  1. For clarity, authors can add to the Supplementary microphotographs of a larger scale to evaluate the morphology of the particles (in addition to Fig. 3).

The additional SEM images of a larger scale were included in the Supplementary Materials (Figure S3).

  1. Have diffraction and morphology analyses been performed on the material after electrochemical testing? This analysis could also profitably complement the above work.

SEM pictures and diffractograms of the electrodes after cycling were added to the Supplementary Materials (Figure S9 and S10, respectively), and a short discussion concerning these investigations was added on Page 12 of the corrected manuscript.

SEM images of the electrode layers before, and after cycling are presented in Figure S9 (in SM). The morphology of the electrode layers of the composites with a lower amount of tin nanoparticles (SiOC/Sn-40% and SiOC:C0.2/Sn-40%) does not change after extended cycling tests (Figure S9. a) and b)), whereas the SiOC:C0.2/Sn-60%) exhibit some cracks (Figure S9. c), which may be attributed to the higher Sn concentration and associated volume changes upon alloying as the tin nanoparticles are accumulated mainly on the ceramic part.

The XRD analysis of cycled samples are presented in Figure S10 (in SM). The diffractograms are quite complex and reveal the presence of graphite, β-Sn (components of the composites), Li2CO3, Sn3O4 (components typical for SEI on tin containing electrodes) [65–67], Li2Sn5 (one of the lowest lithiation states of Li-Sn alloys) and Cu (current collector). The XRD patterns of the cycled binary and ternary composites are similar. The quantitative analysis would be ambiguous due to overlapping of the XRD reflexes and strong peaks coming from the copper current collector.

Reviewer 2 Report

This manuscript reported  a  silicon oxycarbide - graphite – tin nanoparticles
composite for lithium-ion batteries. The results are interesting and conclusions are reliable.

1) While there are some grammar or spelling errors in English. for example, " Final ternary composites were obtaned through ". 

2) The graphs are not clear, should be improved. 

Author Response

This manuscript reported  a  silicon oxycarbide - graphite – tin nanoparticles
composite for lithium-ion batteries. The results are interesting and conclusions are reliable.

Thank you for your valuable comments concerning our manuscript. We corrected the manuscript according to your suggestions.

1) While there are some grammar or spelling errors in English. for example, " Final ternary composites were obtaned through ". 

We carefully revised English in our manuscript and has been checked by a native speaker.

2) The graphs are not clear, should be improved.

We inserted higher-resolution figures in the corrected manuscript.